# An NLP Benchmark Dataset for Assessing Corporate Climate Policy Engagement

**Gaku Morio**[*] and **Christopher D. Manning**
Stanford University
{gaku,manning}@stanford.edu

## Abstract

As societal awareness of climate change grows, corporate climate policy engagements are attracting attention. We propose a dataset to estimate corporate climate policy engagement from various PDF-formatted documents. Our dataset comes from LobbyMap (a platform operated by global think tank InfluenceMap) that provides engagement categories and stances on the documents. To convert the LobbyMap data into the structured dataset, we developed a pipeline using text extraction and OCR. Our contributions are: (i) Building an NLP dataset including 10K documents on corporate climate policy engagement. (ii) Analyzing the properties and challenges of the dataset. (iii) Providing experiments for the dataset using pre-trained language models. The results show that while Longformer outperforms baselines and other pre-trained models, there is still room for significant improvement. We hope our work begins to bridge research on NLP and climate change.

## 1 Introduction

Climate change is one of the most critical challenges confronting our society today [24]. As societal awareness of climate change heightens, how corporations minimize their environmental impact has come under tight scrutiny by the public [12]. Consumers are increasingly warming up to "eco-friendly" products [3] and investors are placing a premium on investments yielding environmental benefits, as evidenced by the popularity of Environment, Social, and Governance (ESG) funds [1].

Nonetheless, skepticism about whether companies that claim environmental benefits are truly effective in mitigating the impact of climate change persists. Controversies have been sparked by dubious practices such as using vague terms to guide consumers to certain irrelevant conclusions (e.g., using 'all-natural' to imply beneficial to the environment) [12] or engaging in lobbying activities to mislead the public and policy-makers [4]. Such practices are often known as *greenwashing* [28, 12, 30, 13] – more formally defined as *"behavior or activities that make people believe that a company is doing more to protect the environment than it really is"*[8]. Greenwashing can help a company to enhance their corporate legitimacy, if it is not exposed [36]. To prevent that from happening, instances of corporate greenwashing have been flagged by environmental, non-profit, and non-governmental organizations. The ongoing public scrutiny deters corporate attempts to mislead stakeholders with deceptive messages [36] and encourages truthful engagement with the public.

Monitoring and identifying greenwashing at scale is difficult as it requires experts with domain knowledge to analyze corporate documents. Natural language processing (NLP) can help to alleviate the issue by automating the process of extracting relevant data quickly. As Stammbach et al. [38] mentioned, the initial step in identifying greenwashing would involve recognizing corporate environmental claims – something that NLP would be well-suited to handle. Previous NLP efforts

---

[*]Also a researcher of Hitachi America, Ltd., Santa Clara, California.

37th Conference on Neural Information Processing Systems (NeurIPS 2023) Track on Datasets and Benchmarks.

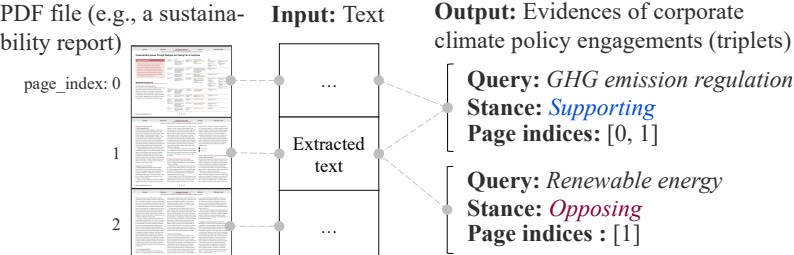

Figure 1: Schematic overview of the proposed task.

in the area include ClimateBERT [43], a domain-adapted language model that incorporates climate-related data, that has been used to label whether a paragraph is climate-related or not [43] and to identify Task Force on Climate-Related Financial Disclosures (TCFD) categories such as governance, strategy and risk management for sentences in corporate reports [6].

However, NLP research in the context of climate change and greenwashing is still in its infancy due to the lack of principles on task definitions and model development. This is primarily due to the lack of open datasets suitable for training and evaluating models within this context. While others have provided datasets in this field [43, 15, 38], they are either narrowly defined, lacking explainability in tasks, not diverse in terms of data source and geography, or small.

Our paper proposes a dataset representing corporations' climate policy engagement collected from a plethora of corporate-related documents that can potentially be used to automatically detect greenwashing. We construct the dataset from LobbyMap [20], a platform operated by global think tank InfluenceMap [18]. As illustrated in Figure 1, our dataset poses a task that takes the text extracted from a PDF file, such as a corporate sustainability report, as input and outputs a set of (query, stance, evidence page indices) triplets. The query represents high-level climate policy issues, such as *'Renewable energy'*. The stance represents a scale with five levels ranging from *'strongly supporting'* to *'opposing.'* The evidence page indices give the supporting pages for the query and stance.

We invested effort in collecting data, and aligning evidence to transform the raw data of LobbyMap into the dataset. Consequently, our dataset differentiates itself from others in terms of its size, label richness, and diversity. Our contributions can be summarized as follows:

**NLP Dataset on Climate Change.** We have assembled a high-quality, large-scale (i.e., including over 10K documents), and diverse dataset, designed for the task of evidence-based assessment of corporate climate policy engagement. We anticipate that our dataset will stimulate research on NLP and climate change, steering it towards more accurate detection of greenwashing.

**Dataset Analysis.** We conducted analyses of the dataset properties, demonstrating that the task of our dataset is challenging as an NLP task.

**Benchmark Experiments.** We evaluated the performance of pre-trained language models such as BERT [14], ClimateBERT, and Longformer [5] on the task. While we obtained promising results, e.g., about 70% F-score for evidence-page detection, there remains ample room for improvement in prediction performance. Furthermore, to establish a benchmark for interpretability and explainability in this task, we introduce and evaluate a supplementary task where the model provides a scrutiny comment for the given query, stance, and evidence page indices.

Overall, we hope that our dataset will stimulate research on NLP and climate change and possibly serves as a foundation for the detection of corporate greenwashing. The code and dataset are available at https://climate-nlp.github.io.

## 2 Related Work

**Background on Greenwashing Controversy.** The term 'greenwashing' was introduced by environmentalist Jay Westerveld in 1986 [12]. The field of greenwashing is extensive and has been the subject of much research in recent years. Some of the existing literature delves into the dynamics between consumers and firms. As the marketplace becomes saturated with an increasing number of "green"

products that tout positive environmental advantages, allegations of greenwashing have concurrently risen [28]. A survey study by de Freitas Netto et al. [12] suggests that certain research highlights *selective disclosure* wherein greenwashing manifests when corporations selectively highlight positive environmental impact. The renowned *seven sins of greenwashing* [39] have been widely discussed and are identified as product-level greenwashing in various studies [12, 13]. For instance, *the sin of the lesser of two evils* pertains to a claim that might be accurate within a specific product category but could potentially divert consumers from the broader environmental implications inherent to that entire category [12].

Nemes et al. [30] contributed a more exhaustive survey, underscoring the gaps in establishing a definition for greenwashing and setting definitive behavioral standards to identify it. A potential connection between our research and their definition of greenwashing might be found in the contexts of *corporate responsibility in action* (where a claim is categorized as greenwashing if it is not mirrored by consistent organizational practices) and *political spin* (deemed as greenwashing when corporations express green undertakings while concurrently lobbying against environmental legislation) [30]. This is because our dataset covers claims from a variety of sources, from official reports to political documents, which can spot potential contradictions in claims.

**Greenwashing-related NLP Research.**

In a few recent years, a burgeoning interest has emerged in leveraging NLP or fact-checking methodologies for the analysis of climate change-centric documents [43, 15]. While not directly addressing greenwashing, there exist Question Answering (QA) systems [10] and bots [34] tailored to facilitate the acquisition of credible climate-related knowledge. These systems typically utilize document retrieval techniques sourcing content from news articles by media and publications disseminated by global institutions to generate responses to user queries. Within the greenwashing context, researchers have implemented keyword analyses in annual reports to analyze mismatches in discourse, actions, and investments [25]. Detecting environmental claims is perceived as a first step towards a greenwashing evaluation [38, 37]. Stammbach et al. [37] introduced a sentence-level classification task for environmental claims in sources such as sustainability reports, earnings calls, and annual reports. Notably, ClimateBERT, a specialized model tailored for the climate domain, has been employed for detecting climate-related paragraphs [43] and for classifying TCFD categories [6]. Bingler et al. [6] provided an analysis that firms are primarily selective in reporting immaterial climate risk information. We believe our research aligns most closely with the detection of environmental claims, while concurrently performing the role of potential fact-checking.

## 3   Understanding LobbyMap

LobbyMap (or its organization InfluenceMap) has been referenced in media outlets [40, 42] as well as in academic research [30], typically in the context of corporate greenwashing. LobbyMap analyzes a wide range of diverse information about companies and industry associations, categorizing each corporation's stance on specific topics. (See also their methodology [21] reproduced in the supplementary material.) LobbyMap features various platform categories, such as regional categories (Japan, United States, South Korea, Australia, and European Union) and a specific corporate group known as the Climate Action 100+ (CA100+) Investor Hub [19], which we focus on in our dataset.

The LobbyMap system consists of several interconnected components, as illustrated in Figure 2. These components work together to offer a comprehensive understanding of a corporation's stance on climate-related issues. Any key terms will be defined and explained in detail later.
**Company Profile.** Each company on LobbyMap has a profile summary page that contains relevant company information, ratings, and a summary review by an expert. These assessments are substantiated by a scoring matrix described below.
**Scoring Matrix.** As illustrated in Figure 2b, the scoring matrix contains 13 *query* rows and 7 *data source* columns. Each cell in the matrix links to a page that contains evidence items pertinent to the selected query and data source.
**Evidence Item.** Each evidence item, as depicted in Figure 2c, records the corporate *stance* for the query along with an excerpt (we call this *evidence snippet*) quoted from an attached data source file in PDF format. This page also includes an analyst's *comment* summarizing the reasons for the assigned stance. One may also obtain other metadata such as the year, creation date, and region associated with the evidence.

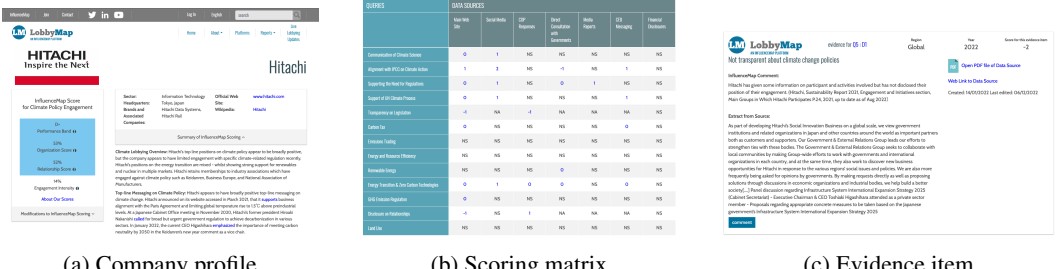

(a) Company profile      (b) Scoring matrix      (c) Evidence item

Figure 2: Example screenshots of LobbyMap pages. The profile page contains a summary of the company, as well as a scoring matrix consists of 13 query labels and 7 data sources. Each matrix cell contains evidence items for the selected query and data source.

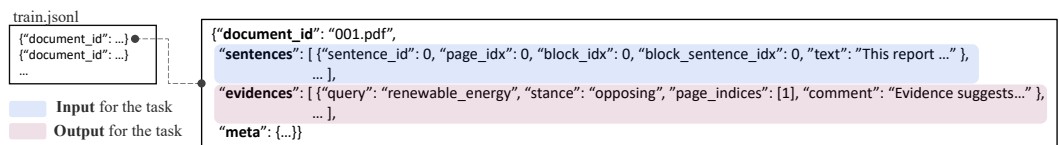

Figure 3: Example data representation of our dataset.

### 3.1 Key Terms in an Evidence Item

**Query.** The query gives the subcategory of the climate policy engagement agenda. For instance, *'Energy transition & zero carbon technologies'* relates to the economy's transition away from carbon-emitting technologies in line with the IPCC's guidelines. A comprehensive list of query definitions can be found in Appendix A.7.

**Data Source.** The data source is characterized by seven distinct document types: *Main Website*, *Social Media*, *CDP Response*, *Direct Consultation with Governments*, *Media Reports*, *CEO Messaging*, and *Financial Disclosures*. PDF files from this data source are attached to the evidence item. Depending on the analyst's approach, a PDF file attached may contain excerpts of only relevant pages, or the entire document. Occasionally, PDF files are screenshots of a website or social media.

**Stance.** A stance is a position expressed on a five-level discrete scale between $+2$ and $-2$. This scale represents whether the company is *'strongly supporting'* ($+2$), *'supporting'* ($+1$), expressing *'no position or mixed position'* (0), *'not supporting'* (-1), or *'opposing'* (-2) the query.

**Evidence Snippet.** An evidence snippet is an excerpt extracted by a human analyst from the attached PDF file in the evidence item. Snippets can be sentences, clauses, or even paragraphs, and may span multiple pages. We use these excerpts to identify the page indices of the PDF file that contain the evidence.

**Comment.** A comment is a brief, human-generated content pertaining to the query, stance, and evidence snippet. It provides an insight into the rationale behind the assigned stance. An example of a comment for a query *'Land use'* is "*Supports forestry sequestration for carbon offsetting but is unclear if supports regulations.*" This comment is used in our supplementary task in Section 6.4.

## 4 Dataset

Our dataset is meticulously curated to offer value to various research purposes. For NLP, this dataset poses a real-world challenge concerning corporate documents in the sustainability domain. For sustainability and finance, models trained on this dataset provide a systematic way to predict a corporate stance on environmental categories.

### 4.1 Dataset Design

Our dataset adheres to two key design principles: First, it should contain all the necessary information to evaluate a corporation's climate policy engagements. Secondly, the data should be stored in a standard format so that it can quickly and easily be used by downstream tasks.

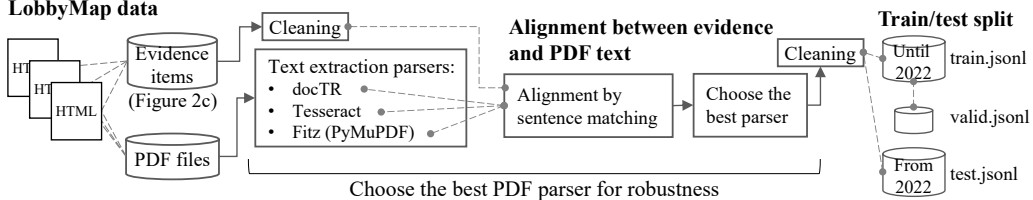

Figure 4: Schematic overview of the dataset construction procedure.

**Information Included in the Dataset.** The dataset meets the first design principle by recording corporate climate policy engagements, which will be used in our task as in Figure 1. Specifically, our dataset will contain text extracted from a document and its associated triplets $(P, Q, S)$, where $Q$, $S$, and $P$ denote a query label, the stance label, and evidence page indices supporting the query and stance, respectively.[2] Our dataset also includes the comment, which is used in the supplementary task described later. The above information provides sufficient information to assess a corporate position on a particular climate issue category.

**Data Representation.** In order to make the dataset easy to use, we store the data in the JSON Lines format since this is widely used in tasks such as fact-verification [41]. Each line contains an instance of our task. An example instance is shown in Figure 3. The *sentences* field contains the text data extracted and sentence-tokenized from the PDF files, including relevant details such as sentence ID and page numbers. Therefore, researchers can directly use the field as task input. The *evidences* field contains information obtained from evidence items (i.e., $P$, $Q$, and $S$.) The *meta* field contains any other metadata, providing origin information for the evidences items.

## 4.2 Dataset Construction

We invested effort in PDF parsing and evidence alignment to transform the raw data of LobbyMap into the dataset. As depicted in Figure 4, the dataset construction primarily consisted of three stages: (i) data collection, (ii) establishing alignments (i.e., correspondences) between the evidence snippet and text in the PDF to identify the evidence page indices, and (iii) splitting data into training, validation, and test sets. Note that we omit pragmatic technique here, but that can be found in Appendix A.5.

**Data Collection.** From February to March 2023, we collected evidence items from companies listed under CA100+ that includes firms key for climate-change. A comprehensive list of these companies can be found in Appendix A.8.

**Text Extraction from PDF Files.** PDF files have varying layouts. In particular, some files contain only embedded images without embedded text. We employed three different PDF text extraction parsers to obtain textual data from the PDF files robustly: Fitz in PyMuPDF [33], docTR [29], and Tesseract [22]. The first is a tool that extracts embedded text from PDF file whereas the last two are Optical Character Recognition (OCR) based software. Our approach of using three parsers to extract textual data minimizes the chances of not obtaining any usable information. We also preserved the order of the text based on the layout produced by each parser.

**Alignment between Evidence Snippet and PDF File.** We need to construct $P$, the indices of the pages containing evidence snippets, because the evidence snippet does not explicitly tell us which part of the PDF file was extracted for the snippet. We used NLTK [7] to split the evidence snippet and text extracted from the PDF file into sentences to align the evidence snippet and PDF at the sentence level. After obtaining alignments from each of the three aforementioned parsers, we selected the one with the highest number of alignment. Finally, we obtained the page indices containing the alignments and designated them as $P$.

**Data Splitting.** We split the data obtained by the previous steps into training, validation and test sets based on the *year* metadata included in the evidence items. Documents before 2022 were assigned into the validation set and train set, while those in or after 2022 were designated as test data. This decision ensures that our model evaluation hinges on more recent data, mirroring a realistic scenario where data from the future is employed for evaluation.

---

[2]The granularity of the *page* is based on pragmatic considerations. See Appendix A.9 for more detail.

Table 1: Basic statistics of the dataset (avg. num. per document in parentheses.)

|  | Train | Validation | Test |
|---|---|---|---|
| # Doc | 7,425 | 825 | 2,354 |
| # Output triplets | 11,159 (1.50) | 1,229 (1.49) | 3,336 (1.42) |
| # Word | 28,434,661 (3829.58) | 3,067,156 (3717.76) | 6,244,125 (2652.56) |
| # Page | 67,091 (9.04) | 7,289 (8.84) | 15,755 (6.69) |

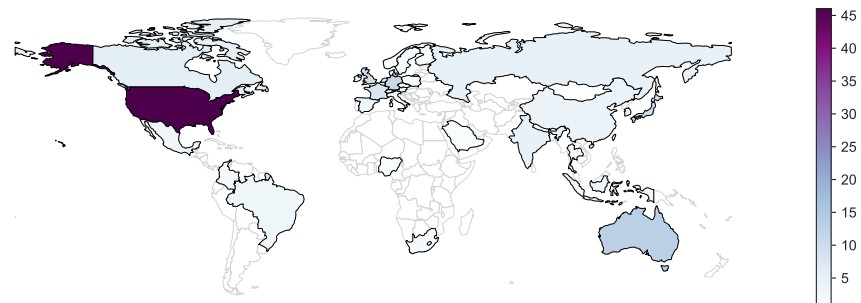

Figure 5: Country distribution of corporate headquarters in our dataset.

## 5 Dataset Analysis

This section provides data analyses and demonstrates the values of the dataset for benchmarking.
**Basic Statistics.** Table 1 shows statistics on the dataset. The training set contains over 7K documents, totalling 28M words distributed across over 60K pages. Also, the number of pages per document is approximately 6-9 and the word counts per document exceed 2,500. This property suggests that our task can be positioned as long document understanding, which is a challenging aspect of NLP. Besides, models must generate multiple triplets per document, increasing the complexity of the task.
**Label Distribution.** We analyzed the label distribution of query and stance. (The full table is in Appendix A.9.1.) We found that the labels are imbalanced. There are about 1K training samples of *'Energy transition & zero carbon technologies'* for *'supporting'*, while the query *'Land use'* shows a sparse presence of labels. The imbalances introduce another level of challenge, where models must predict less frequent labels accurately. We also found a general trend of companies leaning towards a positive stance. This trend may be explained by the fact that companies often seek to enhance their public reputation.
**Diversity.** We found that our dataset is diverse in terms of:
**(i) Time.** The dataset includes evidence items spanning over years. (See Appendix Figure 8.) The increasing number of samples collected each year suggests that the reporting and/or scrutiny have increased over time.
**(ii) Data Source.** The most frequent data source is *Main Web Site* and the least is *Financial Disclosures* (See Appendix Figure 9.) Interestingly, *Social Media* and *CEO Messaging* are also frequent, showing our dataset contains various sources.
**(iii) Geographic Diversity Regarding the Company.** Figure 5 shows the geographical distribution of the corporate headquarters in our dataset. We can see that our dataset contains companies not only from major economic powers such as United States, but also from resource-rich countries such as Australia, which has been accused of state-sponsored greenwashing [17]. The dataset covers all the continents except Antarctica, allowing researchers to perform their analysis across various type of corporations.
**(iv) Geographic Diversity Regarding the Evidence Item Region.** In LobbyMap, each evidence item is associated with a specific regional target. For example, a document from an Australian media article sometimes describe regional events and can be associated within Australia. In the case of a multinational company, it may be associated within 'Global' that is not bound to a specific region. We examined such geographic diversity of evidence items (See Appendix Figure 11 for more details.) We found that 'Global' appears frequently, suggesting most companies in our dataset are multinational. The number of evidence items from Europe and United States is enormous.
**(v) Sector.** There are 14 different types of company sectors in our dataset: automobiles, chemicals, construction materials, consumer staples, energy, food products, healthcare, industrials, information

technology, metals & mining, paper & forest products, retailing, transportation, and utilities. The wide variety of sectors represented in our dataset suggests that it may be of interest to researchers studying diverse industrial fields.

## 6 Benchmark Experiments

Here, we benchmark the proposed task on the dataset. Conceptually, our task projects an input text extracted from a document into a set of output triplets $Y$, where $(P, Q, S) \in Y$.

### 6.1 Models

First, we provide a most-frequent baseline that always outputs majority labels: ($P = \{0\}$, $Q =$ '*Energy transition & zero carbon technologies*', $S =$ '*supporting*') for each document. This is similar to the majority class baseline and is useful as a simple consideration of the task's lower bounds.

Next, given that our benchmark will be used as baselines for future work, we decided to use pre-trained language models that are widely used in current state-of-the-art studies. The challenge of this is that it is difficult to handle dozens of pages at the same time because most pre-trained language models have a limited context length. To this end, we employ a page-wise classification approach, where each document is split into pages, and we feed text of each page into the model and obtain output labels, gathering all the outputs in the document.

We have the following three page-wise classifiers as a pipeline: (i) The *evidence page detector* is fine-tuned to predict whether each page contains evidence or not. (ii) The *query classifier* predicts query labels (i.e., multi-label classification) given a page detected by (i). Gold page indices are used for fine-tuning. (iii) The *stance classifier* predicts one of the five stance labels given the detected page and the predicted query label. Gold page indices and query labels are used for fine-tuning.

As basis of the classifiers, BERT [14] and its variant models, RoBERTa [27], ClimateBERT [43] and Longformer [5], with a classification head, are used. BERT and RoBERTa are known for the strong baseline in text classification tasks. ClimateBERT is pre-trained on sustainability and climate domain so we can verify the effectiveness of domain adaptation. We also provide Longformer, which is specialized for long-document understanding. These pre-trained language models are used as initial weights for the parameters of each of the three classifiers. The optimizer is Adam [23].

For comparison, we introduce a simple linear model using logistic regression [11] and tf-idf [35]. While this model also employs the page-wise approach, it provides a distinct logistic regression binary classifier for each query and stance label.

**Implementation and Hyperparameters.** The input text for each page is created by concatenating all the sentences in the page. During inference for evidence page detection, if none of the pages are identified as containing evidence, the page with the highest probability is considered as the evidence page. For the stance classification, the input text is created by inserting query label in front of the page sentences. If the query classification generates multiple query labels for a single page, we create separate input text for each query label. Finally, we post-process the outputs by gathering evidence page indices which share the same query and stance labels. The implementation detail and hyperparameters are shown in Appendix A.10.

### 6.2 Evaluation Metrics

We evaluate the model outputs in the aspects of evidence page detection, query classification, and stance classification. We provide three types of F-score metrics as follows:
**Strict.** This is a standard F-score, based on the set of tuples produced by a model and gold. The F-score for evidence page indices is evaluated using the set of output tuples and the set of gold tuples. The F-score is calculated for the page indices by singletons $(P)$, for the query by tuples $(P, Q)$ and for the stance by tuples $(P, S)$.
**Page overlap.** The above metric is "too strict" and can not capture how close are the model predictions to gold. To this end, we provide an evidence page overlap-based metric. This is based on the work of Barnes et al. [2], which calculates graph-based structured sentiment F-scores using the ratio of word token overlap between predicted and gold outputs. In this study, the F-scores are calculated by overlap ratio of gold and predicted evidence page indices. Thus, the more overlap, the better the

Table 2: Evaluation results, measuring test F-score (%). $Q$, $S$, and $P$ represent query, stance and evidence page indices, respectively.

| | Document | | | Page overlap | | | Strict | | |
|---|---|---|---|---|---|---|---|---|---|
| | $P$ | $Q$ | $S$ | $P$ | $Q$ | $S$ | $P$ | $Q$ | $S$ |
| Most-frequent | 46.7 | 52.6 | 36.8 | 51.8 | 25.6 | 19.8 | 41.2 | 19.6 | 17.5 |
| Linear | 66.0 | 61.9 | 50.3 | 71.4 | 44.5 | 36.1 | 52.0 | 31.2 | 27.0 |
| BERT-base | 71.0 | 63.5 | 51.6 | 73.6 | 48.1 | 37.2 | 50.2 | 31.9 | 25.8 |
| ClimateBERT | 71.8 | 64.0 | 52.8 | 74.4 | 48.9 | 39.0 | 50.2 | 32.2 | 26.8 |
| RoBERTa-base | 71.6 | 64.5 | 53.1 | 73.8 | 49.6 | 38.3 | 50.4 | 33.4 | 26.6 |
| Longformer-base | 73.7 | 66.9 | 54.6 | 75.9 | 53.0 | 40.8 | 52.5 | 36.1 | 28.6 |
| Longformer-large | 73.9 | 68.8 | 57.3 | 76.5 | 55.0 | 44.1 | 53.6 | 38.7 | 31.5 |

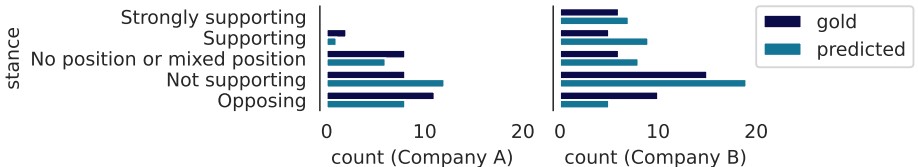

Figure 6: For two companies in the energy sector, the distributions of gold and predicted stances for *'Energy transition & zero carbon technologies'* are shown. The model used is Longformer-large.

F-score is. For query and stance labels scores, the overlap is considered only for cases which share the same query or stance label. For more detail, refer to Appendix A.10.

**Document.** This is the most rough metric where F-scores for query, stance, and evidence page indices are evaluated independently. The motivation of this metric is that there can be cases where the user wants to know just overall trends of query or stance of a company, not fine-grained evidence-based outputs, e.g., one can examine whether energy sector firms are increasingly mentioning renewable energy. For the F-score of evidence page indices, a set of output tuples $s$ and a set of gold tuples $g$, where each tuple represents (document id, $i$) and $i$ denotes an evidence page index, are used to calculate F-score. Similarly, F-score is calculated for query by output tuples (document id, $Q$) and for stance by (document id, $S$). In other words, an output of query/stance is considered correct if the query/stance label is correct, even if the evidence page indices for the output are incorrect.

### 6.3 Result and Discussion

Table 2 provides an overview of the F-score for the test set. For the results of validation set, refer to Appendix Table 8. The results illustrate that all models can detect evidence page indices, with an F-score of about 70% in the document or page overlap $P$ metrics. Given that most PDF files contain only one page (See Appendix Figure 10), this result may be generous. Nevertheless, given the F-score in the document $P$ metric of the most-frequent baseline, which always outputs a page index of 0, shows a low F-score, we can see that training the model is highly effective. On the other hand, F-scores of the strict metric suggests the difficulty of our task to exactly identify evidences. This insight will be further explained later. Query ($Q$) and stance ($S$) classification proves to be challenging. All models demonstrate lower F-scores in these aspects across all the metrics, which might reflect the intricate nature of these tasks.

In terms of pre-trained language models, ClimateBERT outperformed BERT. This indicates that pre-training on the sustainability domain is effective in our task. However, the better trained but not climate-specific RoBERTa partially outperformed ClimateBERT. In turn, Longformer outperformed other models like RoBERTa, showing its robustness in handling long documents. Interestingly, the linear model outperformed BERT in the strict metric for $P$ and $S$. The linear model had higher precision but lower recall than BERT. The strict metric deems predicted page indices incorrect if they deviate at all from the gold indices, potentially disadvantaging high-recall models like BERT. These findings highlight the importance of using multiple metrics, including document and page overlap.

**Error Analysis – Classification.** We investigate representative error patterns of the Longformer-large model in the document metric. For the query, one of the most frequent errors is that the model

predicted *'Energy transition & zero carbon technologies'* and *'Renewable energy'*, while the gold is *'Energy transition & zero carbon technologies.'* This might be because the label *'Energy transition & zero carbon technologies'* sometimes includes the topic of *'Renewable energy'*, and the model could not distinguish that. For the stance, the frequent error is that the model predicted *'supporting'*, while the gold is *'no position or mixed position.'* This suggests that the model finds it difficult to distinguish different nuances regarding the stance.

**Error Analysis – Longer Document.** The lower strict F-score, as noted earlier, seems to be influenced by the length of the documents. Figure 7 illustrates a decline in the prediction performance for $P$ in the Longformer-large model with an increase in the number of pages per document. Inherently, due to the metric's nature, $Q$ and $S$ scores are as low as $P$, as they are constrained by $P$. This indicates that our model struggles with longer documents in terms of the fine-grained (i.e., strict) metric. This issue might be because our model, which operates training and inference on the page-by-page manner, does not consider the context of the entire document.

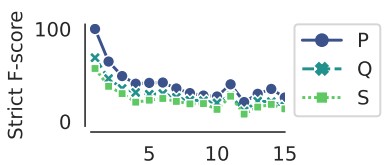

Figure 7: Strict F-score based on the number of pages in the document.

The above results and error analyses suggest the necessity of a domain-adapted model that is capable of comprehending lengthy contexts. Specifically, improving the model's focus on the overall document context and the consistency of multi-label output could potentially enhance its performance.

**Pilot Study.** We present a pilot study aimed at greenwashing detection. As depicted in Figure 6, we select two major companies (A and B) from the energy sector and aggregate the predicted stances towards the query *'Energy transition & zero carbon technologies.'*[3] The figure shows that our predictions are reasonably capable of replicating the trends present in the gold. The important finding here is that the trend of companies A and B is different even in the same sector. The company A tends to present negative stances, whereas the company B exhibits a more ambiguous stance. Company B's presentation of conflicting stances raises the possibility of greenwashing. Even if this were not the case, such comparisons allow researchers to uncover hypotheses worth exploring further. Thus, our dataset can also be used to model and test hypotheses about the corporate climate policy engagement.

## 6.4 Comment Generation: A Supplementary Task for More Explanation

One can recognize that the output triplet $(P, Q, S)$ may not provide sufficient context in some cases, specifically when there is a need to understand how the evidence page is interpreted and how the query and stance are derived from the evidence. Therefore, we introduce a supplementary task: generating comments (as described in Section 3) for a given query, stance, and evidence page indices. We fine-tune FlanT5 [9], where the model is trained by the comments as reference. The generated comments are evaluated by ROUGE [26]. The input text for the model is formatted as *'Generate a reason why the corporate climate policy engagement for "QUERY" is "STANCE". <PAGE TEXTS>'*, where *QUERY* and *STANCE* represent the query and stance label respectively. *<PAGE TEXTS>* represents the concatenated sentences from all reference evidence pages ($P$.)

Table 3: Comment generation evaluation in ROUGE (%.)

|  | R-1 | R-2 | R-L |
| --- | --- | --- | --- |
| Test |  |  |  |
| FlanT5-large | 38.4 | 22.1 | 34.8 |
| FlanT5-XL | 39.5 | 22.7 | 35.6 |
| Validation |  |  |  |
| FlanT5-large | 43.4 | 28.1 | 40.5 |
| FlanT5-XL | 42.7 | 27.6 | 39.7 |

Table 3 represents evaluation results, suggesting the potential of using generative models to generate rationale comments for given triplets $(P, Q, S)$. The FlanT5-XL model outperforms the FlanT5-large model on test data. Performance on the test set is lower than on the validation set. Since the validation and train sets are sampled from the same temporal span, this result suggests that generation performance can be undermined due to changes in the distribution of comments over time.

---

[3]Note that our objective here is not to definitively characterize these companies, but to illustrate the concept of our use case. Thus, we have anonymized the company name.

# 7 Conclusion and Future Work

The field of research aimed at effectively utilizing NLP to contribute to solving climate change issues is still in its infancy. However, endeavors undertaken in recent years have demonstrated the considerable scope for advancement in this realm [38, 43]. We introduced an NLP dataset to predict corporate climate policy engagement. This study is one of the attempts to provide fundamental knowledge in this budding research area. We hope that the proposed dataset will stimulate research on NLP and climate change by laying the foundation for the detection of corporate greenwashing.

We recognize that there is future work to be done. The benchmark experiments revealed room for improvement in prediction performance. We describe several promising directions for future research:
(i) Multi-modal. The development of multi-modal models capable of processing not only text from PDF files but also embedded images could enhance the prediction performance.
(ii) Multi-lingual. While we only considered English text in this work, the capability to work with other languages will allow us to create a more diverse dataset.
(iii) Few-shot learning. By leveraging the capabilities of Large Language Models (LLMs), we can explore strategies for few-shot learning, potentially enabling more efficient training with smaller amounts of data.

## Acknowledgments and Disclosure of Funding

Dylan Tanner, Edward Collins, Chris Hurst, who founded InfuenceMap in 2015, and Harri Rowlands provided valuable data and allowed us to make our datasets publicly available. We thank them for their assistance and interest in the work. Computational resources of AI Bridging Cloud Infrastructure (ABCI) provided by National Institute of Advanced Industrial Science and Technology (AIST) were used for this work. This research was done in the Stanford Data Science (SDS) Affiliates Program. The first author (GM) conducted this study as a Visiting Scholar at Stanford University. GM is also an employee of Hitachi America, Ltd. and received financial support for this study. We thank Chi Heem Wong, Yasushi Miyata, Terufumi Morishita, Arnab Chakrabarti, and the anonymous referees for their helpful comments on this paper.

**About Dataset Use.** The dataset used in this analysis is part of InfluenceMap's content and was used and is released with the latter's approval. InfluenceMap maintains (since 2015) an ongoing database containing millions of data points each consisting of evidence pieces around corporate climate/nature claims and performance. These are scored against globally accepted science based benchmarks such as the IPCC and the IPBES. Subsets of this data for ML/AI and other analysis are available by request from InfluenceMap and use of InfluenceMap's content is subject to Terms and Conditions. Please contact us at info@influencemap.org for more information (kindly use an organizational email).

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

# A  Appendix

## A.1  Ethics Statement

In alignment with the Code of Ethics as outlined by NeurIPS[4], we offer the following discussion on the ethical considerations and potential societal impact of our research.

**Privacy.** Our dataset comprises text gathered from corporate reports, news media, social media posts and so on. While corporate reports and news media may contain company names, product names, and names of executives or employees, these are typically public-facing pieces of information.

**Consent.** Our dataset includes information such as corporate disclosures and news articles, and we recognize that the information are intended to be publicly available.

**Deprecated Datasets.** Not applicable.

**Copyright and Fair Use.** Given that we are using data from LobbyMap, the license for the data itself is copyrighted to the organization.

**Representative Evaluation Practice.** We discuss the geographic diversity of our dataset in our paper, and the sectors of the companies included and their headquarters' locations are listed in Appendix A.8. As a result, we have managed to cover companies from regions, including major continents. However, we must acknowledge that our dataset is predominantly composed of data from companies belonging to economic and resource-rich nations. Users of our dataset should be mindful of this point.

**Safety.** Our research does not employ technologies that directly inflict harm on individuals.

**Security.** Our models can potentially be applied to automate investment decisions and corporate auditing. If the model produces erroneous outputs, or if a user inputs malicious data, it could distort the system's judgment and possibly cause accidents resulting from investment or auditing errors. The labels in our dataset are limited, and the model's recall is not flawless. This implies that models might fail to detect specific environmental claims and stances. Such shortcomings could incentivize firms to continue unjustified activities. Also, although not our intended use, firms can use the model to analyze their own documents and adopt unfair strategies to increase their reputation.

**Discrimination.** Our dataset and models can potentially be used for evaluating corporate climate policy engagement. The model outputs may negatively appraise specific companies, potentially affecting their reputation or that of their employees. In the worst-case scenario, it could contribute to discrimination against specific companies or employees. To minimize such risks, researchers should exercise particular caution when evaluating specific companies and publicizing the results.

**Surveillance.** Our dataset is not targeted at specific individuals but rather at corporations.

**Deception & Harassment.** We believe there is a low likelihood that this dataset could contribute to hate speech or harassment.

**Environment.** Similar to many other machine learning studies, the environmental impact due to energy consumption during model development is recognized. Moreover, while we take the utmost care in creating our dataset, errors in the original data or the dataset creation process could cause the model to make erroneous judgments. Such incorrect decisions could lead to misguided investment decisions or environmental measures, which could consequently impose an environmental burden. Researchers should not apply the dataset or model to decision-making or policy-making.

**Human Rights.** Not applicable.

**Bias and fairness.** The likelihood of our dataset containing biases towards specific races, genders, etc., is low. Biases related to specific countries, regions, or industrial sectors may be considered. Researchers should be aware of the potential existence of bias mainly towards corporations.

## A.2  Limitations

**Data Quality.** We proposed a dataset based on LobbyMap data; although LobbyMap data is generated by expert analysts based on an established methodology, the specific process is unknown and may

---

[4]https://neurips.cc/public/EthicsGuidelines

contain biases that we do not perceive. In addition, since an agreement study is not conducted on the data, we do not know the extent to which the task will achieve consensus among the experts.

In addition, although we have collected data from a wide variety of firms based on CA100+ to construct the dataset, its comprehensiveness remains insufficient given the huge number of firms that exist worldwide. We have not been able to validate its performance for companies other than those included in the dataset.

**Bias.** We recognize that the environmental policy framework within our data might be biased. For instance, a reviewer highlighted that the science of climate change is dynamic; solutions deemed pivotal and popular one year may prove ineffective later. To stay aligned with current information and mitigate this bias, one could either incorporate newer data or diversify data sources. On the other hand, when the model assesses a firm's stance, it's inevitable that the evaluation will reflect the inherent bias of the policy, although the validity of this bias varies depending on the use case.

Bias can also arise from the training data of the pre-trained models. For instance, the policy might favor a specific region or industry.

**Dataset Construction.** We implemented a number of programs based on pragmatic assumptions to construct the dataset. Although we paid utmost attention to the quality of the dataset, these processes may cause the dataset to contain incorrect information. The output results of PDF parser contain a lot of noise, so the input text will contain many errors caused by layout analysis or OCR. In addition, the dataset do not cover all the evidences in LobbyMap, since evidences that either failed to parse a PDF file or failed to make the alignment are dropped from the dataset. For example, evidences where the text is written in Japanese or Chinese would be discarded because the alignment algorithm only considers English text. Also in LobbyMap, depending on the PDF file, it may represent the entire report or only some pages cut out. Consistency in the composition of PDF files is not guaranteed. Therefore, researchers should take into account the possibility that our dataset may contain errors, noises and biases on the task. However, we believe that technological advances in PDF parsing and alignment can address this problem by continually updating the dataset.

LobbyMap only provides documents that contain an evidence item, not documents that do not contain evidences. This means our dataset does not have 'negative samples' in terms of documents. We are based on the strong assumption that at least one page of each document contains evidence item, limiting the benchmark capability for the negative samples.

**Model and Evaluation.** Our proposed pipeline approach performs classification on a page-by-page manner. This does not account for contextual interactions between pages, which may lead to degradation of predictive performance. In addition, our approach cannot be applied to documents that do not contain evidence items because of the above mentioned strong assumption (that at least one page of each document contains an evidence.)

Our models use the text extracted from PDF files as input, while they do not use other modality such as layout, image and table presented in each page. This limits the expressive power of the input, leading to possible predictive performance degradation.

In the supplementary task, we evaluated generative models that output comments given the gold triplets $(P, Q, S)$ as input, but we do not perform end-to-end evaluation. Therefore, using the prediction results of the pipeline approach to generate comments would lead to predictive performance degradation.

### A.3 Computational Resource

For all experiments, we used NVIDIA A100 GPUs provided by ABCI.[5] As of 2022, ABCI ranks 32nd in the Green500 benchmark for power consumption, making it an energy efficient system.[6] We did not conduct hyperparameter search, however, we consumed the computation resource for preliminary experiments. In the experiments, no pre-training was conducted, only fine-tuning, so we believe the environmental impact is considered low.

---

[5]https://docs.abci.ai/en/system-overview/
[6]https://www.top500.org/lists/green500/list/2022/11/

### A.4 Code and Data Availability

The code and data are available at `https://climate-nlp.github.io`.

### A.5 Dataset Construction Detail

Here, we detail the dataset construction procedure.

#### A.5.1 PDF Parsers

In instances where a PDF file reading results in an error, such as I/O errors, we assume that the parser extracts no text. For each PDF parser, we set a maximum limit of 100 pages per document. Any document containing more than 100 pages omits the excess pages. We tokenize the text of each page into sentences using NLTK and follow the text order as per the 'block' outputs of each parser from the PDF file.

#### A.5.2 Evidence Cleaning

To filter out invalid or ambiguous evidence items from the collected LobbyMap data, we applied pragmatic filters. First, we exclude evidence items whose associated companies do not correspond with the company list from CA100+. Second, we remove duplicated evidence items that arose during the data collection process. Third, we discard evidence items that have no attached PDF file or are associated with multiple PDF files, retaining those with a single attached PDF file. Lastly, we remove evidence items that lack evidence excerpts, which were typically indicated as '–no extract–' in the original data. In the final two steps, we eliminate all evidence items referring to PDF files that were referenced by the evidence item to be deleted.

#### A.5.3 Evidence Alignment

During evidence alignment, we apply a partial string match between sentences in the evidence snippet and sentences on each page of a document. We consider pairs with at least a 95% partial string match as aligned, but only for sentences that contain at least five words. Pages with aligned sentences are included in the evidence page indices. However, to filter out alignment noise, we exclude pages containing 14 words or fewer. Furthermore, when we encounter failure in reading PDF data, the associated evidence items are discarded.

#### A.5.4 Resolving Document Duplication

Occasionally, different PDF files contained identical contents even though they were saved under different names. As such, we merged the evidence items for documents with identical hash values of the PDF data that was parsed by each parser.

Also, by calculating the hash value of the all sentences contained in a document, we verified that the input documents for the training data does not leak into that for the test data.

#### A.5.5 Best Parser Selection

If all parsers failed to extract text from a document, we discarded that document. We selected the best parser based on the number of alignments. If the same number of alignments were produced by multiple parsers, we followed a preferential order of Fitz, Tesseract, and docTR.

#### A.5.6 Comment Generation Data

In some cases, multiple evidence items are associated with the same triplet $(P, Q, S)$. The differences in comments between such evidence items often arise from paraphrasing or differing target company names. To create a practical gold standard for comments, we selected the longest comment from among the candidates.

### A.6 Dataset Analysis Detail

Figure 8 shows the year distribution of our dataset.

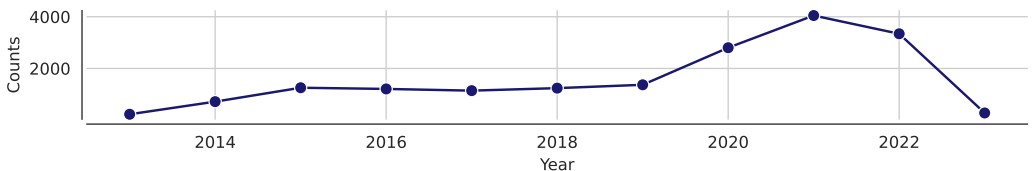

Figure 8: Year distribution of our dataset

Figure 9 shows the data source distribution.

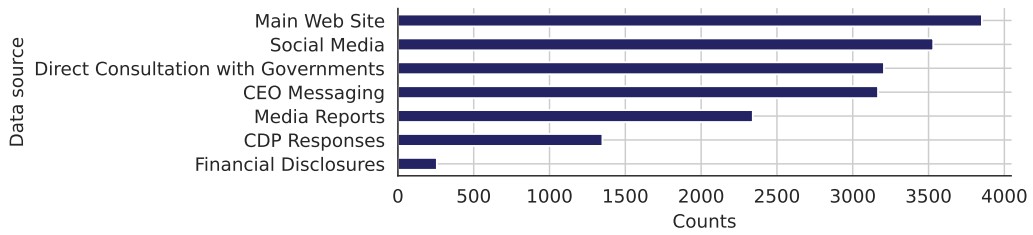

Figure 9: Data source distribution of our dataset

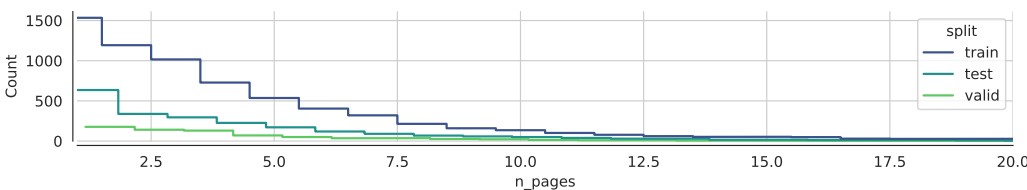

Figure 10: Histogram of page numbers in a document

Figure 10 shows the page number distribution of documents in our dataset. The figure suggests that most of document contain less than 10 pages. In some PDF files, they contain only selected pages that contain evidences rather than entire pages.

Figure 11 shows the evidence item region distribution of selected regions or countries.

### A.7 Full Query List

The following 13 query definitions are cited as is from the scoring matrix page of LobbyMap.

- *Alignment with IPCC on climate action.* Is the organization supporting the science-based response to climate change as set out by the IPCC?
- *Carbon tax.* Is the organisation supporting policy and legislative measures to address climate change: carbon tax.
- *Communication of climate science* Is the organization transparent and clear about its position on climate change science?
- *Disclosure on relationships.* Is the organization transparent about its involvement with industry associations that are influencing climate policy, including the extent to which it is aligned with these groups on climate?
- *Emissions trading.* Is the organisation supporting policy and legislative measures to address climate change: emissions trading.
- *Energy and resource efficiency* Is the organization supporting policy and legislative measures to address climate change: energy efficiency policy, standards, and targets.

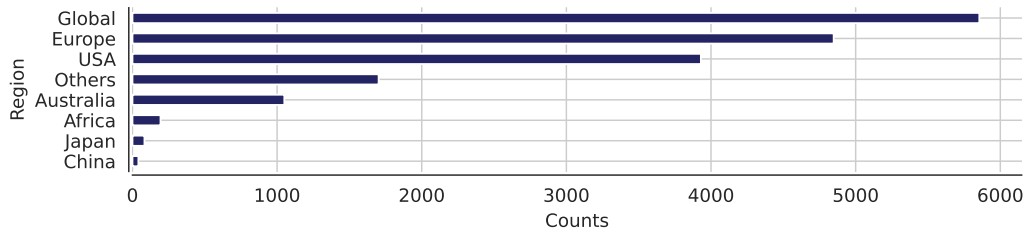

Figure 11: Distribution of selected evidence item regions

- *Energy transition & zero carbon technologies* Is the organization supporting an IPCC-aligned transition of the economy away from carbon-emitting technologies, including supporting relevant policy and legislative measures to enable this transition?
- *GHG emission regulation.* Is the organization supporting policy and legislative measures to address climate change: GHG emission standards and targets. Is the organization supporting policy and legislative measures to address climate change: Standards, targets, and other regulatory measures directly targeting Greenhouse Gas emissions.
- *Land use.* Is the organization supporting policy and legislative measures to enhance and protect ecosystems and land where carbon is being stored?
- *Renewable energy.* Is the organization supporting policy and legislative measures to address climate change: Renewable energy legislation, targets, subsidies, and other policy.
- *Support of UN climate process.* Is the organization supporting the UN FCCC process on climate change?
- *Supporting the need for regulations.* To what extent does the organization express the need for regulatory intervention to resolve the climate crisis?
- *Transparency on legislation.* Is the organisation transparent about its positions on climate change legislation/policy and its activities to influence it?

## A.8 Full Company List

We show the full company list in our dataset below:

| Company | Sector | Headquarters |
| --- | --- | --- |
| Adbri (Adelaide Brighton) | Construction Materials | Adelaide, Australia |
| AGL Australia | Utilities | Sydney, Australia |
| Air France-KLM | Transportation | Paris, France |
| Air Liquide | Chemicals | Paris, France |
| Airbus Group | Industrials | Leiden, Netherlands |
| American Airlines Group | Transportation | Fort Worth, United States |
| American Electric Power | Utilities | Columbus, United States |
| Anglo American | Metals & Mining | London, United Kingdom |
| Antam (Aneka Tambang) | Metals & Mining | Jakarta, Indonesia |
| ArcelorMittal | Metals & Mining | Luxembourg, Luxembourg |
| BASF | Chemicals | Ludwigshafen, Germany |
| BHP | Metals & Mining | Melbourne, Australia |
| Bluescope Steel | Metals & Mining | Melbourne, Australia |
| BMW Group | Automobiles | Munich, Germany |
| BP | Energy | London, United Kingdom |
| Bayer | Healthcare | Leverkusen, Germany |
| Berkshire Hathaway | Industrials | Omaha, United States |
| Boeing | Industrials | Chicago, United States |
| Boral Limited | Metals & Mining | Sydney, Australia |
| Bumi Resources | Metals & Mining | Jakarta, Indonesia |
| Bunge Limited | Consumer Staples | St. Louis, United States |
| CEZ | Utilities | Czech Republic, Czech Republic |
| CNOOC | Energy | Hong Kong, China |
| CRH | Construction Materials | Dublin, Ireland |
| Canadian Natural Resources Ltd (CNRL) | Energy | Calgary, Canada |
| Caterpillar | Industrials | Peoria, United States |
| Grupo Argos | Construction Materials | Medellín, Colombia |
| Cemex | Construction Materials | Monterrey,, Mexico |
| Centrica | Utilities | Berskshire, United Kingdom |
| Chevron | Energy | San Ramon, United States |
| China Petroleum & Chemical Corporation (Sinopec) | Energy | Beijing, China |

| | | |
|---|---|---|
| China Shenhua Energy | Metals & Mining | Beijing, China |
| China Steel | Metals & Mining | Kaohsiung, Taiwan |
| Coal India | Metals & Mining | Kolkata, India |
| Coca-Cola | Consumer Staples | Atlanta, United States |
| Colgate-Palmolive | Consumer Staples | New York, United States |
| ConocoPhillips | Energy | Houston, United States |
| Cummins | Industrials | Columbus, United States |
| Daikin Industries | Industrials | Osaka, Japan |
| Dangote Cement | Construction Materials | Lagos, Nigeria |
| Danone | Food Products | Paris, France |
| Delta Air Lines | Transportation | Atlanta, United States |
| Devon Energy | Energy | Oklahoma City, United States |
| Dominion Energy | Utilities | Richmond, United States |
| Dow Inc | Chemicals | Midland, United States |
| Duke Energy | Utilities | Charlotte, United States |
| E.ON | Utilities | Dusseldorf, Germany |
| EDF | Utilities | Paris, France |
| Eni | Energy | Rome, Italy |
| Ecopetrol | Energy | Bogota, Colombia |
| Enbridge | Energy | Calgary, Canada |
| Enel | Utilities | Rome, Italy |
| Engie | Utilities | Paris, France |
| Eskom Holdings Soc Limited | Utilities | Sunninghill, South Africa |
| Exelon | Utilities | Chicago, United States |
| ExxonMobil | Energy | Irving, United States |
| FirstEnergy Corp | Utilities | Akron, United States |
| Ford Motor | Automobiles | Dearborn, United States |
| Fortum | Utilities | Espoo, Finland |
| Naturgy (Gas Natural Fenosa) | Utilities | Barcelona, Spain |
| Gazprom | Energy | Moscow, Russia |
| General Electric | Industrials | Fairfield, United States |
| General Motors | Automobiles | Detroit, United States |
| Glencore International | Metals & Mining | Baar, Switzerland |
| Grupo México | Metals & Mining | Mexico City, Mexico |
| HeidelbergCement | Construction Materials | Heidelberg, Germany |
| Hitachi | Information Technology | Tokyo, Japan |
| Foxconn | Information Technology | Taipei, Taiwan |
| Honda Motor | Automobiles | Tokyo, Japan |
| Iberdrola | Utilities | Bilbao, Spain |
| Imperial Oil Limited | Energy | Calgary, Canada |
| Incitec Pivot | Chemicals | Southbank, Australia |
| Trane Technologies (formerly Ingersoll-Rand) | Industrials | Davidson, United States |
| International Paper Company | Paper & Forest Products | Memphis, United States |
| ENEOS Holdings (formerly JX Holdings Inc) | Energy | Tokyo, Japan |
| Korea Electric Power Corporation (KEPCO) | Utilities | Naju, South Korea |
| Kinder Morgan | Energy | Houston, United States |
| Philips | Information Technology | Amsterdam, Netherlands |
| Holcim | Construction Materials | Jona, Switzerland |
| Lockheed Martin | Industrials | Bethesda, United States |
| Lukoil | Energy | Moscow, Russia |
| LyondellBasell Industries | Chemicals | Rotterdam, Netherlands |
| Nornickel (MMC Norilsk Nickel) | Metals & Mining | Moscow, Russia |
| Marathon Petroleum | Energy | Houston, United States |
| Martin Marietta Materials | Construction Materials | Raleigh, United States |
| Mercedes-Benz Group | Automobiles | Germany |
| Moller Maersk (Maersk) | Transportation | Copenhagen, Denmark |
| NRG Energy | Utilities | Houston, United States |
| NTPC Ltd | Utilities | New Delhi, India |
| National Grid | Utilities | London, United Kingdom |
| Nestlé | Consumer Staples | Vevey, Switzerland |
| NextEra Energy | Utilities | Juno Beach, United States |
| Nippon Steel Corporation | Metals & Mining | Tokyo, Japan |
| Nissan | Automobiles | Yokohama, Japan |
| OMV | Energy | Vienna, Austria |
| Occidental Petroleum | Energy | Greenway Plaza, United States |
| Oil and Natural Gas Corporation (ONGC) | Energy | India |
| Orica | Industrials | Melbourne, Australia |
| Origin Energy | Energy | Sydney, Australia |
| PACCAR | Automobiles | Washington, United States |
| Pemex (Petróleos Mexicanos) | Energy | Mexico City, Mexico |
| PGE (Polska Grupa Energetyczna) | Utilities | Warsaw, Poland |
| PPL Corporation | Utilities | Allentown, United States |
| PTT Global Chemicals | Chemicals | Bangkok, Thailand |
| Panasonic | Information Technology | Osaka, Japan |
| PepsiCo | Consumer Staples | Harrison, United States |
| PetroChina Company Limited | Energy | Beijing, China |
| Petrobras | Energy | Rio de Janeiro, Brazil |
| Phillips 66 | Energy | Houston, United States |
| POSCO | Metals & Mining | Pohang, South Korea |
| Power Assets Holdings Limited | Utilities | Hong Kong, Hong Kong |
| Procter & Gamble | Consumer Staples | Cincinnati, United States |

| | | |
|---|---|---|
| Qantas Airways | Transportation | Brisbane, Australia |
| RWE | Utilities | Essen, Germany |
| Reliance Industries Limited | Energy | Mumbai, India |
| Renault | Automobiles | Boulogne-Billancourt, France |
| Repsol | Energy | Madrid, Spain |
| Rio Tinto Group | Metals & Mining | London, United Kingdom |
| Rolls-Royce | Industrials | London, United Kingdom |
| Rosneft | Energy | Moscow, Russia |
| Shell | Energy | London, United Kingdom |
| SK Innovation Co | Energy | Seoul, South Korea |
| SSAB | Metals & Mining | Stockholm, Sweden |
| SSE | Utilities | Perth, United Kingdom |
| Saint-Gobain | Construction Materials | Paris, France |
| Santos | Energy | Adelaide, Australia |
| Sasol | Chemicals | Johannesburg, South Africa |
| Saudi Aramco | Energy | Dhahran, Saudi Arabia |
| Severstal | Metals & Mining | Cherepovets, Russia |
| Siemens Energy | Energy | Munich, Germany |
| South32 | Metals & Mining | Perth, Australia |
| Southern Company | Utilities | Atlanta, United States |
| Equinor (formerly Statoil) | Energy | Stavanger, Norway |
| Stellantis | Automobiles | Amsterdam, Netherlands |
| Suncor Energy | Energy | Calgary, Canada |
| Suzano (formerly Fibria Celulose) | Paper & Forest Products | Salvador, Brazil |
| Suzuki | Automobiles | Hamamatsu, Japan |
| Teck Resources Limited | Metals & Mining | Vancouver, Canada |
| AES Corporation | Utilities | Arlington, United States |
| thyssenkrupp | Metals & Mining | Buisburg and Essen, Germany |
| Toray Industries Inc. | Chemicals | Tokyo, Japan |
| TotalEnergies | Energy | Paris, France |
| Toyota Motor | Automobiles | Toyota City, Japan |
| TC Energy | Energy | Calgary, Canada |
| UltraTech Cement | Construction Materials | Mumbai, India |
| Unilever | Consumer Staples | London, United Kingdom |
| Uniper | Energy | Düsseldorf, Germany |
| United Airlines | Transportation | Chicago, United States |
| Raytheon Technologies Corporation (formerly United Technologies) | Industrials | Hartford, United States |
| United Tractors | Industrials | Jakarta, Indonesia |
| Vale | Metals & Mining | Rio de Janeiro, Brazil |
| Valero Energy | Energy | San Antonio, United States |
| Vedanta Resources | Metals & Mining | London, United Kingdom |
| Vistra Corp | Utilities | Irving, United States |
| Volkswagen Group | Automobiles | Wolfsburg, Germany |
| Volvo Group | Automobiles | Gothenburg, Sweden |
| WEC Energy Group Inc | Utilities | Milwaukee, United States |
| Woolworths Ltd | Consumer Staples | Bella Vista, Australia |
| Walmart Stores | Retailing | Bentonville, United States |
| Weyerhaeuser Company | Paper & Forest Products | Seattle, United States |
| Woodside | Energy | Perth, Australia |
| XCEL Energy | Utilities | Minneapolis, United States |

Table 5: Query and stance distribution

| | Strongly supp. | | | Supp. | | | No or mixed pos. | | | Not supp. | | | Opposing | | |
|---|---|---|---|---|---|---|---|---|---|---|---|---|---|---|---|
| | train | valid | test | train | valid | test | train | valid | test | train | valid | test | train | valid | test |
| Alignment with IPCC on climate action | 409 | 54 | 161 | 387 | 38 | 85 | 221 | 29 | 62 | 171 | 21 | 11 | 12 | 0 | 0 |
| Carbon tax | 53 | 9 | 5 | 159 | 21 | 28 | 84 | 16 | 19 | 115 | 7 | 17 | 86 | 6 | 1 |
| Communication of climate science | 180 | 29 | 57 | 206 | 21 | 71 | 19 | 1 | 2 | 24 | 5 | 1 | 17 | 1 | 0 |
| Disclosure on relationships | 28 | 0 | 2 | 74 | 7 | 9 | 63 | 7 | 13 | 136 | 16 | 23 | 23 | 1 | 2 |
| Emissions trading | 169 | 22 | 10 | 210 | 26 | 27 | 130 | 10 | 25 | 182 | 18 | 35 | 47 | 6 | 9 |
| Energy and resource efficiency | 102 | 17 | 29 | 145 | 17 | 46 | 125 | 13 | 29 | 87 | 15 | 15 | 85 | 4 | 2 |
| Energy transition & zero carbon technologies | 286 | 24 | 129 | 1205 | 137 | 497 | 622 | 78 | 353 | 1035 | 91 | 343 | 526 | 61 | 222 |
| GHG emission regulation | 267 | 29 | 40 | 220 | 22 | 67 | 221 | 25 | 82 | 217 | 25 | 36 | 157 | 9 | 20 |
| Land use | 8 | 0 | 1 | 18 | 1 | 16 | 39 | 2 | 20 | 2 | 1 | 2 | 0 | 0 | 0 |
| Renewable energy | 150 | 16 | 49 | 172 | 23 | 76 | 123 | 16 | 57 | 166 | 14 | 30 | 203 | 22 | 13 |
| Support of UN climate process | 165 | 16 | 9 | 433 | 60 | 143 | 55 | 4 | 10 | 18 | 6 | 0 | 6 | 0 | 0 |
| Supporting the need for regulations | 85 | 7 | 23 | 317 | 33 | 114 | 229 | 24 | 94 | 197 | 22 | 45 | 6 | 0 | 2 |
| Transparency on legislation | 13 | 3 | 3 | 65 | 4 | 12 | 58 | 4 | 8 | 99 | 12 | 17 | 27 | 1 | 7 |

Table 6: Hyperparameters

| | BERT-base | ClimateBERT | Longformer-base | Longformer-large | RoBERTa-base |
|---|---|---|---|---|---|
| Learning rate | 1e-5 | 1e-5 | 1e-5 | 1e-5 | 1e-5 |
| Batch size | 8 | 8 | 8 | 8 | 8 |
| Training steps | 20000 | 20000 | 20000 | 20000 | 20000 |
| Warmup ratio | 0.1 | 0.1 | 0.1 | 0.1 | 0.1 |
| Max sequence len. at training | 512 | 512 | 1532 | 1532 | 512 |
| Max sequence len. at inference | 512 | 512 | 2048 | 2048 | 512 |

## A.9 Page Consideration

Although "evidence snipet" is extracted by a human, it is sometimes not consistent in terms of granularity; it can be consists of phrases, sentences, or paragraphs. Thus, we thought training and evaluating models on the origin granularity is not always suitable. Moreover, extracting sentence or paragraph from the PDF file depends on PDF parsers. This means, if one uses different a parser, the output labels of the dataset will need to be reconstructed. Our dataset is considered to be used for benchmarking, so the compatibility is important. Therefore, we believe page-level evidence detection is a pragmatic approach.

However, we have included information of sentence-level evidence alignments in the metadata so that researchers can refer to that.

### A.9.1 Query and Stance Distribution

Table 5 shows the complete table describing the distribution of query and stance labels.

## A.10 Implementation and Hyperparameters

We used PyTorch [31] and HuggingFace [44] libraries for the model implementation.

For the pipeline approach, we applied post-processing technique where evidence page indices are merged for output evidences that have the same query and stance labels.

For the linear model, we used scikit-learn [32] and its default hyperparameters.

For the overlap metric, we calculate the weighted true positive based on page overlap. Precision is defined an $TP_{\text{weighted}}/(TP + FP)$. For example, given a gold tuple $(P, Q) = ((0, 1), \text{"renewable\_energy"})$ and a predicted tuple $(P, Q) = ((1, 2), \text{"renewable\_energy"})$, $TP = 1$ since at least one page index overlaps between the gold and predicted tuples. $TP_{\text{weighted}}$ becomes $1/2$ because, among the gold page indices $(0, 1)$, only one page index (i.e., $1$) overlaps with the predicted indices. Recall is calculated in the same notion.

We show hyperparameters used in our experiments in Table 6 and Table 7. We did not conduct any hyperparameter search and model selection by the validation set. Since our goal is not to achieve the best performance, but to compare models, we used fixed hyperparameter values commonly used in other studies across experiments.

Table 7: Hyperparameters used in the supplementary task

|  | FlanT5-large | FlanT5-XL |
|---|---|---|
| Learning rate | 5e-4 | 5e-4 |
| Batch size | 32 | 32 |
| Training epochs | 10 | 10 |
| Warmup ratio | 0.1 | 0.1 |
| Max sequence len. at training | 1532 | 1532 |
| Max sequence len. at inference | 2048 | 2048 |
| Min sequence len. at inference | 150 | 150 |
| Beam size | 3 | 3 |

Table 8: Evaluation results in validation F-score (%). $Q$, $S$, and $P$ represent query, stance and evidence page indices, respectively

|  | Document | | | Page overlap | | | Strict | | |
|---|---|---|---|---|---|---|---|---|---|
|  | $P$ | $Q$ | $S$ | $P$ | $Q$ | $S$ | $P$ | $Q$ | $S$ |
| Most-frequent | 44.8 | 37.5 | 35.4 | 49.0 | 19.1 | 16.8 | 40.3 | 15.5 | 14.7 |
| Linear | 68.1 | 55.8 | 50.7 | 72.2 | 41.9 | 37.6 | 53.3 | 30.4 | 27.8 |
| BERT-base | 70.6 | 58.7 | 51.7 | 72.8 | 45.2 | 39.4 | 52.8 | 32.7 | 28.7 |
| ClimateBERT | 71.4 | 60.7 | 53.0 | 73.9 | 47.5 | 39.3 | 52.9 | 33.2 | 28.3 |
| RoBERTa-base | 71.6 | 62.5 | 55.3 | 73.7 | 48.3 | 41.2 | 52.5 | 34.1 | 30.2 |
| Longformer-base | 73.4 | 62.4 | 58.3 | 74.8 | 49.8 | 44.3 | 53.2 | 35.1 | 31.9 |
| Longformer-large | 75.2 | 65.9 | 57.6 | 76.7 | 53.7 | 46.2 | 55.0 | 38.3 | 33.2 |

## A.11 Evaluation Results

Table 8 shows the evaluation results for the validation set.

## A.12 Dataset Documentation

We follow the existing dataset sheets [16] to provide information of our dataset with the dataset consumers.

### A.12.1 Motivation

**For what purpose was the dataset created?**    This dataset is designed for the task of evidence-based assessment of corporate climate policy engagement. The task takes the text extracted from a PDF file, such as a corporate sustainability report, as input and outputs a set of (query, stance, evidence page indices) triplets. We anticipate that our dataset will stimulate research on NLP and climate change, steering it towards more accurate detection of greenwashing.

**Who created the dataset (for example, which team, research group) and on behalf of which entity (for example, company, institution, organization)?**    The authors of this paper created the dataset. GM is a Visiting Scholar at Stanford University, and this work is supervised by CDM at the Stanford NLP Group.

**Who funded the creation of the dataset?**    This research was done in the Stanford Data Science (SDS) Affiliates Program.

### A.12.2 Composition

**What do the instances that comprise the dataset represent (for example, documents, photos, people, countries)?**    The dataset represents documents, e.g., corporate-related reports and screenshots of web media. The origin document could contain not only text but images.

**How many instances are there in total (of each type, if appropriate)?**    See Table 1 and Table 5.

**Does the dataset contain all possible instances or is it a sample (not necessarily random) of instances from a larger set?**    Many samples are removed when the dataset construction system failed to parse PDF files or found any issues. All samples are retained except the those that have been technically removed.

**What data does each instance consist of?**    Our dataset mainly consists of text data extracted from a PDF file, evidences (i.e., $(P, Q, S)$ triplets and its comment text.) The dataset also contains metadata (e.g., origin third party data and PDF parser name.) Also, we provide company information and origin PDF data.

**Is there a label or target associated with each instance?**    Yes. See Section 4.

**Is any information missing from individual instances?**    N/A.

**Are relationships between individual instances made explicit (for example, users' movie ratings, social network links)?**    The PDF file name is assumed as an ID in our dataset, and each instance can be resolved mainly by the PDF file names.

**Are there recommended data splits (for example, training, development/validation, testing)?**    We recommend using our split described in the paper.

**Are there any errors, sources of noise, or redundancies in the dataset?**    Yes, there are. See Appendix A.2.

**Is the dataset self-contained, or does it link to or otherwise rely on external resources (for example, websites, tweets, other datasets)?**    The dataset is self-contained for the tasks described in the paper; links for the origin web pages are included to clarify the source. a) We can not ensure the linked resources exist for a long time and consistent. b) There is no official archive related to the linked resource.

**Does the dataset contain data that might be considered confidential (for example, data that is protected by legal privilege or by doctor–patient confidentiality, data that includes the content of individuals' non-public communications)?** No.

**Does the dataset contain data that, if viewed directly, might be offensive, insulting, threatening, or might otherwise cause anxiety?** No.

**Does the dataset identify any subpopulations (for example, by age, gender)?** No.

**Is it possible to identify individuals (that is, one or more natural persons), either directly or indirectly (that is, in combination with other data) from the dataset?** No.

**Does the dataset contain data that might be considered sensitive in any way (for example, data that reveals race or ethnic origins, sexual orientations, religious beliefs, political opinions or union memberships, or locations; financial or health data; biometric or genetic data; forms of government identification, such as social security numbers; criminal history)?** No.

### A.12.3 Collection process

**How was the data associated with each instance acquired? Was the data directly observable (for example, raw text, movie ratings), reported by subjects (for example, survey responses), or indirectly inferred/ derived from other data (for example, part-of-speech tags, model-based guesses for age or language)?** See Section 4.

**What mechanisms or procedures were used to collect the data (for example, hardware apparatuses or sensors, manual human curation, software programs, software APIs)?** See Section 4 and Appendix A.5.

**If the dataset is a sample from a larger set, what was the sampling strategy (for example, deterministic, probabilistic with specific sampling probabilities)?** No probabilistic sampling has been done. We removed items that could not be technically included in the dataset.

**Who was involved in the data collection process (for example, students, crowdworkers, contractors) and how were they compensated (for example, how much were crowdworkers paid)?** N/A.

**Over what timeframe was the data collected?** The data collection was conducted during February to March 2023. Each evidence items have its own timeframe (i.e., creation date or update date.)

**Were any ethical review processes conducted (for example, by an institutional review board)?** No.

**Did you collect the data from the individuals in question directly, or obtain it via third parties or other sources (for example, websites)?** Via third party websites.

**Were the individuals in question notified about the data collection?** N/A.

**Did the individuals in question consent to the collection and use of their data?** N/A.

**If consent was obtained, were the consenting individuals provided with a mechanism to revoke their consent in the future or for certain uses?** N/A.

**Has an analysis of the potential impact of the dataset and its use on data subjects (for example, a data protection impact analysis) been conducted?** N/A.

### A.12.4 Preprocessing/cleaning/labeling

**Was any preprocessing/cleaning/labeling of the data done (for example, discretization or bucketing, tokenization, part-of-speech tagging, SIFT feature extraction, removal of instances, processing of missing values)?** Yes. See Section 4 and Appendix A.5.

**Was the "raw" data saved in addition to the preprocessed/cleaned/ labeled data (for example, to support unanticipated future uses)?** Yes, origin evidence item information can be found in the meta field in the dataset.

**Is the software that was used to preprocess/clean/label the data available?** Not yet available.

### A.12.5 Uses

**Has the dataset been used for any tasks already?** No.

**Is there a repository that links to any or all papers or systems that use the dataset?** See Appendix A.4.

**What (other) tasks could the dataset be used for?** Any form using the dataset can be considered, such as multi-modal climate policy detection task and evidence generation task.

**Is there anything about the composition of the dataset or the way it was collected and preprocessed/ cleaned/labeled that might impact future uses?** See Appendix A.1.

**Are there tasks for which the dataset should not be used?** See Appendix A.1.

### A.12.6 Distribution

**Will the dataset be distributed to third parties outside of the entity (for example, company, institution, organization) on behalf of which the dataset was created?** Yes.

**How will the dataset be distributed (for example, tarball on website, API, GitHub)?** See Appendix A.4.

**When will the dataset be distributed?** After this paper is accepted, as soon as possible.

**Will the dataset be distributed under a copyright or other intellectual property (IP) license, and/or under applicable terms of use (ToU)?** See Appendix A.1. Except for third-party works, the extent to which they are attributed to the authors is distributed under specific license that can be found under the dataset repository.

**Have any third parties imposed IP-based or other restrictions on the data associated with the instances?** The third party data is not for commercial use and is otherwise subject to the terms and conditions of the organization.

**Do any export controls or other regulatory restrictions apply to the dataset or to individual instances?** No.

### A.12.7 Maintenance

**Who will be supporting/hosting/maintaining the dataset?** The authors will be.

**How can the owner/curator/ manager of the dataset be contacted (for example, email address)?** By the email address.

**Is there an erratum?** N/A.

**Will the dataset be updated (for example, to correct labeling errors, add new instances, delete instances)?** It is possible that the dataset be updated on the website or codebase to correct errors or to delete instances upon requests.

**If the dataset relates to people, are there applicable limits on the retention of the data associated with the instances (for example, were the individuals in question told that their data would be retained for a fixed period of time and then deleted)?** N/A.

**Will older versions of the dataset continue to be supported/hosted/ maintained?** It depends on the nature of the dataset update. We may notify the dataset consumers via a website or codebase.

**If others want to extend/augment/build on/contribute to the dataset, is there a mechanism for them to do so?** No. We consider that the dataset is used only for our task. Any extended work should be a separate contribution with ours.

