# OpenReview forum: "An NLP Benchmark Dataset for Assessing Corporate Climate Policy Engagement"
_NeurIPS.cc/2023/Track/Datasets_and_Benchmarks — NeurIPS 2023 Datasets and Benchmarks Spotlight_

### Official Review · Reviewer_mNPN · 2023-07-20
**Corporate Climate Policy Benchmark**

**Rating:** 8
**Confidence:** 3
**Clarity:** The paper is well written and I did n…

**Strengths:**

Corporate produced documents are driven by public-relations concerns and are likely to be amendable to automated processing and likely to  be of interest to researchers in both academic and journalistic settings.
I could also see philanthropic organizations and maybe even investors expressing interest in using this dataset.

**Additional Feedback:**

I think this work is quite unique, and the aggregation of large numbers of corporate policy documents might have other applications beyond environmental policy monitoring. Perhaps this proposal would be stronger if the set of policies included some other policies issues that are less potentially polarized. For example, support for philanthropic activities, or conversations in corporate documents regarding tax policies. Having invested in producing the data already, it doesn't seem like that much of a lift to consider other aspects (the authors are much better positioned to think of these) as a way to validate the methodology. I would expect that companies are more similar, especially within industries, when compare on issues where there's less PR incentives to distinguish their brand. The degree to which each company is "green" is largely driven by PR goals, and that makes it difficult to control in an experimental setting.

**Correctness:**

The dataset was reasonably constructed, and the benchmark criteria as well seem both reasonable and sufficiently challenging.

**Documentation:**

The appendix describes the PDF's from the companies that comprise the dataset as being public domain. I don't think this is correct, and it is entirely possible that some corporations may seek to have their content removed from the dataset. However, I expect that a broad fair-use case can be made here.

Otherwise, I think the documentation on how the data was processed is exemplary.

**Ethics:**

I think this work is principled, and likely valuable to the public, but the authors in their ethics statement give a worse case scenario of companies or their executives being harassed by the disclosure of the their greenwashing activities through this benchmark. I would suggest that another equally potentially harmful case would be the failure to detect environmental campaigns and for companies to continue to avoid consequences. (This is the very nature of pollution where companies externalize their costs to the public.) This potential outcome seems not to have been thought of by the authors.

**Limitations:**

The authors detail several of the limitations and potential consequences that data processing steps may have on the quality of the resulting dataset. I think these potential problems are real, but likely minor.

From the overall scope, I don't think the authors have considered the potential risks associated with formalizing and automating of the detection of greenwashing. Companies are strongly motivated and well funded when it comes to establishing a chosen perception of their programs, and are known to engage in these practices. I have no doubt that companies would closely study any algorithmic system used to evaluate them and seek to game this for advantage.

**Opportunities For Improvement:**

Focusing exclusively on corporate documents probably leaves large gaps in the potential for understanding. While this data spans years, the science on climate change is always evolving, and solutions that seem popular and important one year can later turn out to be ineffective. The troubling developments around so-called carbon offsets have made them less appealing (and the negative publicity associated with the poor stewardship of some carbon-offset programs has made them perhaps toxic for some companies.) I have concerns that the policy issues framework (which is not spelled out in the paper itself) may contain biases. I'm also concerned that the language models as well may convey biases regarding certain environmental policies based on the training material.

**Relation To Prior Work:**

There's no related work, and unfortunately there's no context regarding the history of misrepresentation in corporate documents. The _Merchants of Doubt_ book comes to mind, but I'm certain there's ample scholarship on this topic that could at least be summarized briefly as it is relevant to explain the nature of the task and why it will be so challenging.

**Summary And Contributions:**

This work collects a highly heterogeneous set of documents from multiple corporations in PDF format and processes them to produce aggregate data on the various companies' policies and statements regarding environmental policies and actions.

---

> ### Author Response · Authors · 2023-08-22
>
> Thank you for your interest in our paper and for raising the discussion.
>
> > Focusing exclusively on corporate documents probably leaves large gaps in the potential for understanding. While this data spans years, the science on climate change is always evolving, and solutions that seem popular and important one year can later turn out to be ineffective. The troubling developments around so-called carbon offsets have made them less appealing (and the negative publicity associated with the poor stewardship of some carbon-offset programs has made them perhaps toxic for some companies.) I have concerns that the policy issues framework (which is not spelled out in the paper itself) may contain biases.
>
> We agree that the framework of the environmental policy contained in data may contain bias. This bias may or may not be correct depending on the use case. If we want to align with more up-to-date information, one possible way to avoid the bias is to use newer data or to cover a variety of data sources. Also, as the model judges a firm's stance, it is inevitable that it will be based on a policy that contains a specific bias. We expect our study to be applied to the development of other policies and labels. We have added this limitation to the revised version.
>
> > I'm also concerned that the language models as well may convey biases regarding certain environmental policies based on the training material.
>
> We agree with your concern. For example, it is possible that the policy may be biased toward a particular region and industry. It is an interesting research direction to investigate how the performance of a model may be affected by biases in a pre-trained model. We have added this limitation to the revised version.
>
> > The authors detail several of the limitations and potential consequences that data processing steps may have on the quality of the resulting dataset. I think these potential problems are real, but likely minor.
>
> We have now evaluated the quality of the OCR text and the quality of the evidence alignment. Please see our response to the reviewer JKh5.
>
> > From the overall scope, I don't think the authors have considered the potential risks associated with formalizing and automating of the detection of greenwashing. Companies are strongly motivated and well funded when it comes to establishing a chosen perception of their programs, and are known to engage in these practices. I have no doubt that companies would closely study any algorithmic system used to evaluate them and seek to game this for advantage.
>
> We understand your concerns. We have now added this perspective to the Ethics Statement in the revised version.
>
> > There's no related work, and unfortunately there's no context regarding the history of misrepresentation in corporate documents.
>
> We have now added the related work section in the revised version.
>
> > The appendix describes the PDF's from the companies that comprise the dataset as being public domain. I don't think this is correct
>
> Thank you for pointing this out. We have removed the wording.
>
> > I would suggest that another equally potentially harmful case would be the failure to detect environmental campaigns and for companies to continue to avoid consequences. (This is the very nature of pollution where companies externalize their costs to the public.) This potential outcome seems not to have been thought of by the authors.
>
> We understand your concerns. The labels of our dataset are limited and the recall of the model output is not perfect. We have added this point to the Ethics Statement.

---

> > ### Comment · Reviewer_mNPN · 2023-08-29
> >
> > It's surprising how much a related work section can enhance a paper - the authors addressed my concerns and placed this into a context both in terms of the history environmental policy and of NLP processing as well as provided context for the small but important ways these overlap. With this context the work here seems both stronger and more relevant so I've raised my score.

---

### Official Review · Reviewer_nhwc · 2023-07-20
**A very good paper, fine for Track Datasets and Benchmarks.**

**Rating:** 9
**Confidence:** 4
**Correctness:** correct
**Clarity:** fine

**Strengths:**

- Created a new NLP dataset including 10K documents on corporate climate policy engagement.
- Analyzed the properties and challenges of the dataset.
- Reported experiments with pre-trained language models on the dataset.
- well-structured and well-written paper.


**Additional Feedback:**

-

**Documentation:**

yes, there is sufficient detail

**Ethics:**

Not.

**Limitations:**

Adequately addressed in the paper.

**Opportunities For Improvement:**

Not really,

**Relation To Prior Work:**

yes

**Summary And Contributions:**

А new dataset to estimate corporate climate policy engagement from various PDF-formatted documents. The dataset comes from LobbyMap, a platform that provides engagement categories and stances on the documents.

---

> ### Author Response · Authors · 2023-08-22
>
> Thank you for your comment.

---

### Official Review · Reviewer_kg8y · 2023-07-22
**Review for the Corporate Climate Change Engagement Benchmark**

**Rating:** 7
**Confidence:** 3
**Clarity:** The paper is well written.

**Strengths:**

* The paper focuses on a problem NLP hasn't helped much yet.
* The benchmark includes PLMs handling not only short but also long texts.

**Additional Feedback:**

L10: double > double the performance? Same comment in L209, about the lower bounds of the task.

L39: “such NLP models” might confuse the reader in the starting sentence of this paragraph

L50 (102-104): Is this a 5-point Likert scale? Why is "strongly opposing” excluded?

L74: “(See also the methodology [19].)” Can you please provide some high-level description of their methodology? I couldn't find much in their website, but even if I did, this information is subject to change over time.

L80: Emphasising in italics, to present later, confused me as a reader. I respect the authors' choice, but also sharing my experience.

L124: How do you assess the quality of text extraction?

Table 5: In four cases, opposing was excluded during testing, making the results of the experiments less trusted. Why did you choose to add all twelve opposing instances of “Alignment with IPCC on climate action” in the training set? Controlling just the year (L160) doesn’t sound right based on this observation.

L157: How is the "degree alignment" defined and what are its statistics (e.g., min, mean, max)?

L160-163: Please clarify that there isn’t any data leakage (e.g., the same information about a company being present in all subsets). From a check of the data, I could not find anything alarming, but it would be best if you explicitly stated this in the text.

L216-218: Perhaps the authors would like to consider renaming to “detector” and “classifier” for clarity.

L218: “the gold page indices are used for fine-tuning” is repeated in L220. Also, please clarify if fine-tuning regards all tasks, as this was not clear to me from this passage.

L226: Fine-tune by Adam> please consider re-wording.

L239 (strict): For the page number, F1 is used as in Dice (intersection / union), applied on the whole evaluation set in one computation. If my understanding is correct, and exact match is assumed per tuple, then if a model has made a mistake by 10 pages is weighted equally to one by 1. An error-based metric could address that. The same applies for S, to avoid ignoring the ordered nature of the scale.

L241: The (inline) math don’t look good to me; perhaps they could have been completely avoided given that precision, recall, and F-score are well known.

L242: I think that the document ID could have been excluded, unless there was a usage that I missed.

L244: In the point scale of S, an error from strongly support to support is less important than if it was to opposed.

L247-251: This is more important to define, compared to the standard precision and recall. Please provide the formulas and perhaps an example (showcasing that the measure above is strict).

L254: What kind of cases are these? Please provide an example.

L257: Evaluating a metric doesn't sound right.

Figure 7: The horizontal scale is different between A and B, making the comparison difficult.

L272: “even” —> why did you expect otherwise?

L291: I understand that labels are tuples, where P is the first element. If the ID is misclassified, then the information of support/opposed is meaningless. If that is correct, couldn't a 2-stage system be assessed, first P, then Q/S?

L321: Regarding the potential, how is Rouge 39.5 interpreted in your context? Also, why ROUGE? Why not a BERT-based score, for example?

**Correctness:**

* The ordered nature of two tasks (S, P) was disregarded, which can be fixed by adding a complementary metric.
* The selected data split introduces imbalance, in some cases even removing completely a class during testing (please read my feedback regarding Table 5).

**Documentation:**

Yes

**Limitations:**

* The authors correctly pointed out the limitation of the data quality (A2). However, they could also provide an assessment, by annotating, for example, a sample, counting and assessing the text extraction errors.
* Using Longformer was well-thought, but the motivation for using BERT is not clear. Given that the documents comprise much more than 512 tokens, BERT could have been applied in a better way (e.g., hierarchically). In any case, though, I agree that simple baselines should not be overlooked, so I suggest that the authors include also a simpler TFIDF baseline (not limited by the text's length), such as SVM.

**Opportunities For Improvement:**

* The text quality is not assessed. Extracted texts of low quality may be included (A2), but there isn't any estimation of how many and how likely that is.
* F1 was used for evaluation for all three topics, suggesting three versions, from strict to lenient. Although an applicable measure, the ordered nature of the labels for two tasks, make this choice questionable. Stance (S) was practically rated from opposed to strongly support while evidence (P) used page numbers. For both these tasks an error-based metric could be added, so that high-error predictions are not equally weighted with low-error ones.
* BERT was used as a baseline, but these are long documents and it is not expected to work out of the box. An alternative would be to use BERT to represent a longer portion of the document (i.e., extracting representations sequentially, up to a limit), then feeding a classification network. Another alternative would be to employ hierarchical models. In any case, I would also suggest a simple SVM with TFIDF features (see also https://arxiv.org/pdf/2306.07111.pdf).
* The possibility of greenwashing is raised from conflicting stances (L304), but it is not clear to me if this is an established assumption. This is important, because it can directly lead to automated detection (L45). In the same line of thoughts, the pilot study (L296) could regard cases where greenwashing is known, to explore the trends.

**Relation To Prior Work:**

The authors discuss the related datasets (L41) and models (L35-38) in the introduction.

**Summary And Contributions:**

This paper presented a dataset and a benchmark related to corporate climate policy engagement, which can be used for monitoring and detecting greenwashing. Texts in English were extracted from corporate-related documents. Snippets were then labeled according to the topic and the stance. Three tasks were benchmarked: detection of the indices of the pages comprising the snippet (P), detection of the topic of the snippet (Q), and estimation of the stance of the company for that topic (S). Results showed that a Longformer, which can handle longer texts, outperforms BERT-based models. A use-case for two anonymised companies showed that the stance trends can be captured across topics, despite the small deviations per topic.

---

> ### Author Response · Authors · 2023-08-22
>
> Thank you so much for the detailed and thorough comments and suggestions. We revised the paper based on your feedback.
> Here are our thought regarding your comments and how we addressed them.
>
> > The text quality is not assessed. Extracted texts of low quality may be included (A2), but there isn't any estimation of how many and how likely that is.
>
> In terms of OCR errors, we have now performed a small study. Please see our response to the reviewer JKh5.
>
> > F1 was used for evaluation for all three topics, suggesting three versions, from strict to lenient. .... For both these tasks an error-based metric could be added, so that high-error predictions are not equally weighted with low-error ones.
>
> While we preliminary considered to provide error-based metrics, we do not believe they are appropriate for the following reasons:
>
> First, introducing error-based metrics for page indices requires the assumption that a page and another page close to it in a document contain semantically close content. Such an assumption could be made for some documents, we do not it is appropriate for many documents. For example, a sustainability report may have text on an entirely different topic when one crosses the page.
>
> Second, we find it difficult to apply error-based metrics to stances as well. One reason is that we are not sure that the 5-point scale evaluated by LobbyMap's own benchmarks is continuous and uniform. For example, it is not obvious that incorrectly predicting mixed_position as supporting and incorrectly predicting mixed_position as not_supporting are equal errors. We also believe that our metric is feasible given that accuracy is used in benchmark datasets with a 5-scaled label such as SST-5.
>
> > BERT was used as a baseline, but these are long documents and it is not expected to work out of the box. .... In any case, I would also suggest a simple SVM with TFIDF features ....
>
> We added experimental results using logistic regression with tfidf. For results and discussion, please refer to the revised paper.
>
> > The possibility of greenwashing is raised from conflicting stances (L304), but it is not clear to me if this is an established assumption.
>
> We believe this assumption is based on recent research. We believe the type of greenwashing to which this pilot study potentially relates includes "political spin" and "corporate responsibility in action" (*Nemes et al., see related study in the revised edition). These two types highlight policy inconsistencies within the organization and lobbying activities that are inconsistent with environmental interests.
>
> As a real example of an energy company by the LobbyMap analysis, despite supporting the need to reduce emissions in line with the 2 degrees Celsius target in the sustainability report (i.e., supporting Alignment with IPCC on Climate Action), in a form filed by the firm with the U.S. SEC, the firm suggests that a net-zero emissions pathway of IEA would lead to a "degradation in global standard of living”" (i.e., not supporting Alignment with IPCC on Climate Action). This case may be greenwashing because of the conflict between the statement to investors and the political activity, according to the two types mentioned above. We believe that our pilot study is a very preliminary analysis to detect this type of greenwash.
>
> [*] Noémi Nemes et al. 2022. An integrated framework to assess greenwashing. Sustainability.
>
> > In the same line of thoughts, the pilot study (L296) could regard cases where greenwashing is known, to explore the trends.
>
> Yes. It is in our future interest to carefully analyze in-depth trends in industries that are accused of greenwashing.
>
> > Table 5: In four cases, opposing was excluded during testing, making the results of the experiments less trusted. .... Controlling just the year (L160) doesn’t sound right based on this observation.
>
> This is a limitation of our dataset. It is possible to adjust the data split so that the opposing stance of specific query labels are included in the test data, but we believe this is not an essential solution, as it would result in an evaluation with a very small sample anyway.
>
> > L291: I understand that labels are tuples, where P is the first element. .... If that is correct, couldn't a 2-stage system be assessed, first P, then Q/S?
>
> We provide an option in our implementation to classify queries and stances using gold page indices. One can use that functionality as needed.
>
> > L321: Regarding the potential, how is Rouge 39.5 interpreted in your context? Also, why ROUGE? Why not a BERT-based score, for example?
>
> The ROUGE score was higher than we expected. This is likely due to LobbyMap's control over the comments being in a particular format. ROUGE was used because it is a lightweight metric and is standardized in document summarization and story generation. Future research can evaluate using a variety of metrics including BERT-based scores.

---

> > ### Comment · Reviewer_kg8y · 2023-08-25
> >
> > I would like to thank the authors for their analytical response and the shared real example. I still find F1 for stances weak, but the explanation about the page indices is convincing. Overall, my concerns are addressed and I would like to see this paper published (increasing my score from 6).

---

### Official Review · Reviewer_JKh5 · 2023-07-31
**A Benchmark for Assessing Corporate Climate Policy Engagement**

**Rating:** 6
**Confidence:** 2
**Correctness:** N/A
**Clarity:** In general, this paper is well writte…

**Strengths:**

1. The field of research aimed at effectively utilizing NLP to contribute to solving climate change issues is limited. The proposed dataset could stimulate research for assessing corporate climate policy engagement with NLP techniques.
2. The whole dataset construction pipeline is straightforward and the presentation is clear.
3. Detailed statistics of the proposed dataset are provided. Five pre-trained language models are evaluated on the proposed dataset.

**Additional Feedback:**

Not applicable.

**Documentation:**

The documentation of how to use the code and dataset is not provided. More details about how the errors in the text extraction from PDFs and alignments between evidence snippets and PDFs affect the quality of the resulting dataset should be provided.

**Ethics:**

This paper provides a detailed ethical discussion in the A.1 Ethical Statements section in Appendix.

**Opportunities For Improvement:**

1. My main concern is the quality of the proposed dataset. The whole construction pipeline is automatic without any human annotations or filtering. Text extracted from PDFs relies on OCR-based tools. The extracted texts could contain lots of noise since existing OCR-based tools may be desirable. Statistics or human analysis about the accuracy of text extraction should be provided to justify the quality of the proposed dataset. Similarly, the alignment between the evidence snippets and the PDF file suffers from the same issue.
2. This paper did not include any sections to discuss how this work differs from previous efforts. For example, previous works (datasets and methods) in the climate policy domain, stance detection, and text classification.
3. More baselines could be included. For example, methods do not use pre-trained language models, and large language models.

**Relation To Prior Work:**

As mentioned above, this paper did not include any sections to discuss how this work differs from previous efforts.

**Summary And Contributions:**

This paper constructs a dataset that consists of 10K documents on corporate climate policy engagement. The dataset is constructed based on LobbyMap, a platform that provides engagement categories and stances on the documents. A pipeline is developed to convert the data from LobbyMap to structural data by using off-the-shelf tools to extract texts from PDFs. Experiments and analysis with pre-trained language models (e.g. BERT, RoBERTa, Longformer) are conducted on the proposed dataset.

---

> ### Author Response · Authors · 2023-08-22
>
> I appreciate your comments and suggestions for improvement. We addressed your concerns as much as we can. Please see below.
>
> > My main concern is the quality of the proposed dataset. The whole construction pipeline is automatic without any human annotations or filtering. Text extracted from PDFs relies on OCR-based tools. The extracted texts could contain lots of noise since existing OCR-based tools may be desirable. Statistics or human analysis about the accuracy of text extraction should be provided to justify the quality of the proposed dataset. Similarly, the alignment between the evidence snippets and the PDF file suffers from the same issue.
>
> We recognize that there can be noise in terms of OCR and evidence alignment in our dataset. However, these effects are minimized by the best efforts achievable with current technology.
>
> #### OCR
>
> More than 80% of all PDF documents have embedded text, so OCR is performed for the limited number of documents. Nevertheless, noise in OCR is our concern. We have now performed a small study of OCR errors. We tested whether the sentences contained OCR errors for the 45 sentences contained in three randomly selected documents that do not have embedded text. The resulting accuracy was 39/45 = 86.7%. Errors include misidentification of "0" and "o". Higher accuracy would be achieved at the character level. We believe that the accuracy of OCR is relatively high because most of the documents are not photographs or scanned documents, which would reduce the accuracy of OCR, but rather screenshots of web pages.
>
> #### Evidence alignment
>
> We believe identifying page indices is reasonably accurate as we have calculated a string match of over 95% to produce an alignment between PDF pages and evidence snippet. We selected 43 evidence items from randomly selected 20 documents with multiple pages and evaluate the alignment accuracy of page indices. The results showed that the page indices were correctly identified with an accuracy of 42/43=97.7%. E.g., a page index is incorrectly detected on a page with similar text of evidence snippet. However, this case can be considered correct in a broader sense because the text has the same meaning.
>
>
> > This paper did not include any sections to discuss how this work differs from previous efforts. For example, previous works (datasets and methods) in the climate policy domain, stance detection, and text classification.
>
> We have now added a related work section in the revised paper.
>
> > More baselines could be included. For example, methods do not use pre-trained language models, and large language models.
>
> We have now added experimental results using logistic regression with tfidf. For results and discussion, please refer to the revised paper.
>
> > The documentation of how to use the code and dataset is not provided.
>
> We will provide codebase and webpage to describe the usage upon acceptance.

---

> > ### Comment · Reviewer_JKh5 · 2023-08-26
> >
> > Thanks for the clarifications. I appreciate that the authors include the related work section and more experimental results in the revised version. The authors address many of my concerns. I would like to increase my rating to 6.

---

### Author Response · Authors · 2023-08-22

Dear reviewers,

Thank you so much for the comments and suggestions on our paper.
By addressing the concerns as much as we can, we have now revised the paper. Red text means additions have been made; blue text means revisions have been made. Other corrections of citation and grammatical errors are included. Also, Table 8 in the Appendix has now been replaced with the validation scores because the test set scores were incorrectly shown.

Please refer to our comments to individual reviewers for specific details.

Best regards,

The authors.

---

### Decision · Program_Chairs · 2023-09-22

**Decision:**

Accept (Spotlight)

**Comment:**

Strengths:
* A useful NLP dataset including 10K documents to help climate change
* The properties and challenges of the dataset are clear
* Includes experiments with pre-trained language models on the dataset.
* Well-structured and well-written paper. Improved a lot during the discussion phase.

Weaknesses:
* Some concerns remain about the quality of the proposed dataset. Especially about potential noise in the extraction process.
* More baselines could be included. For example, methods that do not use pre-trained language models, and large language models.

Overall a high-quality paper with potentially significant impact.